# Multi-Probiotic *Lactobacillus* Supplementation Improves Liver Function and Reduces Cholesterol Levels in Jeju Native Pigs

**DOI:** 10.3390/ani11082309

**Published:** 2021-08-05

**Authors:** Dahye Kim, Yunhui Min, Jiwon Yang, Yunji Heo, Mangeun Kim, Chang-Gi Hur, Sang-Chul Lee, Hak-Kyo Lee, Ki-Duk Song, Jaeyoung Heo, Young-Ok Son, Dong-Sun Lee

**Affiliations:** 1Department of Animal Biotechnology, Faculty of Biotechnology, College of Applied Life Sciences, Jeju National University, Jeju-si 63243, Korea; dahyekim@jejunu.ac.kr (D.K.); yangjw1127@naver.com (J.Y.); dbswl6363@jejunu.ac.kr (Y.H.); aksxhs123@jejunu.ac.kr (M.K.); 2Interdisciplinary Graduate Program in Advanced Convergence Technology and Science, Jeju National University, Jeju-si 63243, Korea; reinise4011@jejunu.ac.kr; 3Cronex Co., Jeju-si 63078, Korea; cghur@cronex.co.kr; 4Cronex Co., Cheongju-si, Chungcheongbuk-do 28174, Korea; dreamporc@cronex.co.kr; 5Department of Animal Biotechnology, Chonbuk National University, Jeonju-si 54896, Korea; breedlee@jbnu.ac.kr (H.-K.L.); peterdamian@naver.com (K.-D.S.); jyheo@jbnu.ac.kr (J.H.); 6Bio-Health Materials Core-Facility Center, Jeju National University, Jeju-si 63243, Korea; 7Practical Translational Research Center, Jeju National University, Jeju-si 63243, Korea; 8Faculty of Biotechnology, College of Applied Life Sciences, Jeju National University, Jeju-si 63243, Korea

**Keywords:** probiotic, *Lactobacillus*, liver function, cholesterol, adipokine

## Abstract

**Simple Summary:**

Probiotics are used in the food industry as feed additives to maintain the balance of animal gut microbiota. They are also considered to have potential therapeutic effects against liver diseases. This study showed that dietary *Lactobacillus* supplementation improved liver function and reduced cholesterol levels in Jeju native pigs, with Toll-like receptors (TLR) signaling as the primary response in the gut against *Lactobacillus* and TNF-α/IFN-γ as the central mediator cytokines in the gut and liver tissues. *Lactobacillus* supplementation may be applied to treat metabolic disorders of the liver, especially cholesterol-related disorders, in farm animals.

**Abstract:**

We evaluated the dietary effects of multiple probiotics in Jeju native pigs, using basal diet and multi-probiotic *Lactobacillus* (basal diet with 1% multi-probiotics) treatments (*n* = 9 each) for 3 months. We analyzed growth performance, feed efficiency, backfat thickness, blood parameters, hematological profiles, adipokines, and immune-related cytokines in pig tissues. Average daily gain, feed intake, feed efficiency, backfat thickness, and body weight were not significantly different between both groups. In *Lactobacillus* group, total protein (*p* < 0.08) and bilirubin (*p* < 0.03) concentrations increased; blood urea nitrogen (*p* < 0.08), alkaline phosphatase (*p* < 0.08), and gamma-glutamyltransferase (*p* < 0.08) activities decreased. *Lactobacillus* group showed decreased adiponectin (*p* < 0.05), chemerin (*p* < 0.05), and visfatin expression in adipose tissues, and increased TLR4 (*p* < 0.05), MYD88 (*p* < 0.05), TNF-α (*p* < 0.001), and IFN-γ (*p* < 0.001) expression in the liver. Additionally, NOD1 (*p* < 0.05), NOD2 (*p* < 0.01), and MYD88 (*p* < 0.05) mRNA levels in proximal colon tissue upregulated significantly. Colon, longissimus dorsi muscle, fat tissue, and liver histological analyses revealed no significant differences between the groups. Conclusively, *Lactobacillus* supplementation improved liver function and reduced cholesterol levels. Its application may treat metabolic liver disorders, especially cholesterol-related disorders.

## 1. Introduction

Probiotics are beneficial microorganisms that compete with harmful intestinal microflora in a host [1]. They have a remarkable ability to alter the intestinal microbiota, stimulate intestinal immunity, improve resistance to diseases, reduce pathogen shedding, and improve health status [2]. Therefore, the application of probiotics on human health and farm animal production has been widely investigated [3]. Primarily, probiotics are widely used in the food industry as feed additives to maintain the balance of animal gut microbiota [4]. *Lactobacillus* species are the most common probiotic agents [5]. *Lactobacillus* feeding has been proposed to improve growth performance, gut microbiota modulation, and immune response; however, these beneficial effects are strain-dependent [6]. It has recently been reported that *Lactobacillus fermentum* MCC2760 and *Lactobacillus paracasei* DTA81 reduce cholesterol levels in a host by exerting substantial bile salt hydrolase activities [7]. In addition, a systemic review reported that *Lactobacillus* diets reduce body weight (BW) and fat mass in overweight human adults [8]. Probiotics are also considered to have potential therapeutic effects against liver diseases [9]. The liver has various important functions such as detoxification, metabolism, immune system, protein synthesis, cholesterol and bile production, micronutrient storage, blood sugar balance, etc.; therefore, liver failure results in high mortality [10].

Improving the intestinal epithelial barrier and immune system is the primary function of probiotics [11]. Toll-like receptors (TLR) play a significant role in the defense mechanism of innate immunity and the induction of adaptive immunity [12]. Soluble proteins of myeloid differentiation 88 (MYD88) are essential adaptor proteins in the TLR signaling pathway. The TLR receptor plays a crucial role in MYD88 signal transduction [13]. The nucleotide-binding oligomerization domain (NOD) is a microbial sensor that can recognize microbes and resist microbial pathogens [14]. NOD regulates intestinal flora and intestinal homeostasis [14]. The NOD receptor acts as a host defense mechanism via NOD1 and NOD2 signals and activates nuclear factor kappa B (NF-κB) and mitogen-activated protein kinase, causing the release of immune response-related cytokines [15]. As an endocrine organ, the adipose tissue can influence all other organs via the release of adipokines, such as adiponectin, chemerin, leptin, and visfatin. These adipokines cause metabolic dysregulation and cardiometabolic diseases [16]. 

Jeju native pigs (JNPs) are mini pigs native to Jeju Island in South Korea. Although Korean consumers favor the meat of JNPs owing to its excellent tenderness, marbling, and juiciness, the production efficiency is low due to its small size, slow growth rate, and low feed consumption [17]. To overcome this limitation and promote the JNPs industry, Korean animal scientists are developing animal models for various pharmaceutical tests or xenotransplantation.

In this study, we aimed to investigate the cholesterol-lowering and liver function-improving properties of multi-probiotic *Lactobacillus* in JNPs. The cholesterol-lowering properties of *Lactobacillus* were examined based on blood cholesterol level, and its effect on liver function was determined based on blood urea nitrogen (BUN), alkaline phosphatase (ALP), and gamma-glutamyltransferase (GGT) levels. We also measured immune response-related cytokines and adipokines.

## 2. Materials and Methods

### 2.1. Animals and Experimental Procedures

All animal experiments were performed in a contract research organization facility at Cronex Co., Ltd. (Cheongju, Korea). Female JNPs (*Sus scrofa* domestica) aged 3 months with an initial mean BW of 32.69 ± 3.37 kg were used to evaluate the effects of dietary multi-probiotic feeding. The animals were randomly divided into two groups (*n* = 9, each group) to have similar mean BW (32.63 vs. 32.75 kg) and were fed one of the two dietary treatments for a feeding period of 3 months. The multi-probiotic was purchased from SunBio (Gunpo-Si, Gyeonggi-do, South Korea). The experimental treatments are as follows: basal diet and *Lactobacillus* diet (basal diet with 1% multi-probiotic (*Lactobacillus buchneri* NLRI-1201; 1.2 × 10^8^ CFU/g, *Lactobacillus plantarum* NLRI-101; 1.6 × 10^8^ CFU/g, *Lactobacillus casei* DK128; 1.4 × 10^8^ CFU/g with fructo-oligosaccharide from *Saccharum barberi* (Samyang Corporation, Seongnam-Si, South Korea). Additionally, a basal group diet based on maize, wheat, soybean meal, and wheat bran was formulated to meet or exceed the nutrient requirements (Appendix A).

### 2.2. Measurement of Growth Performance and Backfat Thickness

To evaluate growth performance in both groups, the pigs were weighed monthly during the experimental period. Backfat thickness was determined approximately 20 cm away from the mid-back positions by physical palpation to accurately ascertain the scanning sites using ultrasonic-type probes (Anyscan BF; SENGKANG GLC, Seongnam-si, Korea).

### 2.3. Blood Collection and Analysis

For biochemical analysis, blood was collected via the jugular vein monthly during the experimental period. Sodium-heparinized vacuum tubes (BD Vacutainer; BD Biosciences, Franklin Lakes, NJ, USA) were used to collect plasma and whole blood. After blood collection, the blood samples were incubated at room temperature for 1 h and centrifuged at 1700× *g* for 30 min to separate the blood plasma and blood cells from the whole blood samples. A Catalyst DX* Chemistry Analyzer and Chem 17 clip (IDEXX Laboratories, Inc., Westbrook, Maine, USA) were used to analyze the blood plasma biochemistry profiles. Complete blood cell count was measured using a ProCyte DX Hematology Analyzer (IDEXX Laboratories, Inc., Westbrook, ME, USA) according to the manufacturer’s instructions. Plasma immunoglobulin G (IgG) levels were quantified using porcine IgG enzyme-linked immunosorbent assay (ELISA) kits (Solarbio Life Sciences & Technology Co., Ltd., Beijing, China).

### 2.4. Tissues Harvesting and Sample Preparation

At the end of the feeding period, pigs aged 6 months were slaughtered. *Longissimus dorsi* muscle, backfat, liver, and proximal colon samples were collected immediately after slaughter. To analyze mRNA and protein expression levels in the tissues, the samples were flash-frozen in liquid nitrogen and stored at −80 °C. For histological analysis, the collected tissues were fixed in 4% paraformaldehyde overnight at 4 °C.

### 2.5. Quantification of Gene Expression Using RT-PCR and qRT-PCR

Total RNA was extracted from *longissimus dorsi* muscle, adipose, liver, and proximal colon tissues using TRIzol reagent (Waltham, MA, USA). The final RNA was eluted in an appropriate amount of 0.1% diethylpyrocarbonate-treated water (Sigma-Aldrich, Inc., St. Louis, MO, USA), and the total RNA concentration was determined using a NanoDrop™ 2000 spectrophotometer (Thermo Scientific, Waltham, MA, USA). For cDNA preparation, total RNA was reverse-transcribed using ImProm-II™ Reverse Transcriptase reagent (Promega, Madison, WI, USA). The expression levels of the target mRNAs were amplified by PCR using SYBR premix ExTaq reagents (TaKaRa Bio, Mountain View, CA, USA) and CFX96™ Real-Time System (BIO-RAD, Hercules, CA, USA) in the Bio-Health Materials Core-Facility, Jeju National University. The specific primers used for amplifying the representative genes are listed in Appendix A. Gene expression levels were calculated using the 2^−ΔΔCT^ method. The mRNA levels of the tested genes were normalized to that of glyceraldehyde-3-phosphate dehydrogenase (GAPDH).

### 2.6. Western Blotting Analysis

The protein content of each tissue was quantified by BCA assay, and 20 μg of protein from each sample was separated on a 10–12% Bis-Tris gel (Bio-Rad Laboratories, Inc., Hercules, CA, USA). The proteins were then transferred to a nitrocellulose blotting membrane (GE Healthcare, NJ, USA), blocked with 5% skim milk (BD Difco, Franklin Lakes, NJ, USA), and incubated overnight at 4 °C with the following primary antibodies: GAPDH, adiponectin, chemerin, and TNF-α (Santa Cruz Biotechnology Inc., Santa Cruz, CA, USA); nampt (Invitrogen, Waltham, MA, USA); TLR4, MYD88, IL-1β, IL-6, and IF-γ (Cell Signaling Technology, Inc., Danvers, MA, USA); and NOD1 and NOD2 (Abcam, Cambridge, UK). The blots were then probed with horseradish peroxidase-labeled secondary antibodies (Solarbio Life Sciences & Technology Co., Ltd., Beijing, China) for 60 min at room temperature and visualized by chemiluminescence using a SuperSignal West Dura Extended Duration Substrate kit (Thermo Scientific). The intensity of the immunoreactive bands was detected using a chemiluminescence imaging system (Bio-Rad Laboratories, Inc., Hercules, CA, USA).

### 2.7. Histological Analysis

Each sample was placed in a histology cassette, dehydrated through graded ethanol to xylene, and embedded in paraffin blocks following standard histological protocols. The blocks were cut at a thickness of 5 μm and processed for standard hematoxylin (Mayer’s Hematoxylin; Dako North America, Inc., Carpinteria, CA, USA) and eosin (Eosin Y; Sigma-Aldrich, St. Louis, MO, USA) staining.

### 2.8. Statistical Analysis

All data in each figure are presented as the mean ± standard error of the mean. Differences between the two groups were statistically analyzed using Student’s *t*-test. Significant differences between the basal diet and *Lactobacillus* groups are indicated as * *p* < 0.05, ** *p* < 0.01, and *** *p* < 0.001. All statistical analyses were performed using IBM SPSS Statistics 24 (International Business Machines Corporation, Armonk, NY, USA).

## 3. Results

### 3.1. Growth Performance

During the feeding period, the BW of pigs in each group increased from 32.75 to 60.38 kg and 32.63 to 56.50 kg for the basal diet and *Lactobacillus* groups, respectively (Table 1). The apparent growth performance for daily gain, feed efficiency, and backfat thickness did not differ significantly between the two groups. The average daily gain throughout the feeding period was 306.94 and 265.28 g/d for the basal diet and *Lactobacillus* groups, respectively. The animals were allotted to the two diets at equal feed intake (g/d), and feed efficiency during the 3 months of feeding was estimated to be 191.84 for the basal diet group and 164.16 for the *Lactobacillus* group, without significant differences. The backfat thickness increased from 7.75 to 14.00 mm for the basal diet group and 8.75 to 14.75 mm for the *Lactobacillus* group. These results suggest that the modulation of gut microbiota by *Lactobacillus* may result in BW or fat loss.

### 3.2. Plasma Biochemical Parameters, Hematological Profiles and Plasma IgG Level

Blood parameters were investigated to confirm the health and nutritional status at the start and end of the experiment. Results of blood plasma biochemical analyses are presented in Table 2 and Appendix A. There was no difference in plasma glucose, creatinine, inorganic phosphate, and calcium concentrations at the start and end of the experiment in both groups. BUN concentration was significantly lower in the *Lactobacillus* group than in the basal diet group (*p* = 0.08). Total protein (*p* = 0.08) and total bilirubin (*p* = 0.03) concentrations were higher in the *Lactobacillus* group than in the basal diet group, whereas cholesterol level (*p* = 0.02), ALP (*p* = 0.08), and GGT (*p* = 0.08) activities were reduced in the *Lactobacillus* group compared with those in the basal diet group. The remaining plasma biochemical parameters did not differ between the two groups. Hematological analysis revealed that the number of neutrophils in the *Lactobacillus* group was higher than that in the basal diet group (*p* = 0.07) (Appendix A). However, the other hematological parameters were not different between the basal diet and *Lactobacillus* groups. Plasma IgG levels increased significantly in response to the multi-probiotic *Lactobacillus* feed (Figure 1). These results suggest that *Lactobacillus* supplementation may alter cholesterol and liver metabolism and immune responses.

### 3.3. Adipokine Expression in Adipose Tissues

We determined the mRNA and protein expression levels of adipokines in pig adipose tissues by PCR and western blotting, respectively. We noted that the mRNA expression of adiponectin (*p* < 0.05), chemerin (*p* < 0.05), and visfatin in adipose tissues was reduced in the *Lactobacillus* group compared with that in the basal diet group (Figure 2). Moreover, the protein levels of adiponectin, chemerin, and visfatin were reduced in the *Lactobacillus* group compared with those in the basal diet group. This pattern of adipokine expression may explain the backfat thickness results.

### 3.4. Expression of TLR4, NOD, and Immune Response-Related Cytokines in the Liver

TLR4 and MYD88 mRNA expression significantly increased in the liver of pigs in the *Lactobacillus* group compared with that in the basal diet group (*p* < 0.05) (Figure 3). The mRNA levels of NOD1 and NOD2 in the liver also increased in pigs fed *Lactobacillus* compared with those in pigs fed basal diet, but the difference was not significant. Similarly, the protein expression of TLR4 and MYD88 in the liver was upregulated in response to *Lactobacillus*. Conventional PCR results showed that the mRNA level of NOD1 in the liver of pigs in the *Lactobacillus* group was slightly upregulated compared with that in the basal diet group, but the increase was not significant (Figure 3e). However, immunoblot results showed more robust upregulation of NOD1 expression in the liver of pigs in the *Lactobacillus* group, compared with that in the basal diet group. In addition, the mRNA and protein levels of the immune response-related cytokines IL-1β, TNF-α (*p* < 0.001), IL-6, and IFN-γ (*p* < 0.001) in the liver tissue were increased in the *Lactobacillus* group compared with those in the basal diet group (Figure 4). These results suggest that TLR4 signaling-mediated expression of TNF-α/IFN-γ is related to the response of the liver to *Lactobacillus*.

### 3.5. TLR4 and NOD Expression in Proximal Colon Tissues

We determined the expression of TLR4 and NOD in the proximal colon tissue in both groups. The mRNA levels of NOD1 (*p* < 0.05), NOD2 (*p* < 0.01), and MYD88 (*p* < 0.05) in colon tissues were significantly upregulated in response to *Lactobacillus* feeding compared with the basal diet (Figure 5). There were no significant changes in TLR4 expression between the groups. TLR4 and NOD signaling changes were further confirmed at the protein level by immunoblotting; these results were significantly consistent with the mRNA analysis results. Furthermore, the expression of immune response-related cytokines in the proximal colon tissue was not altered, except for IL-1β expression (*p* < 0.05), which was significantly decreased in the *Lactobacillus* group compared with that in the basal diet group (Appendix A).

### 3.6. Histological Changes in the Proximal Colon, Muscle, Adipose, and Liver Tissues

We determined the effect of multi-probiotic *Lactobacillus* feeding on histological changes in the proximal colon, *longissimus dorsi* muscle, adipose, and liver tissues of each group using hematoxylin and eosin staining. Histology analysis revealed that the proximal colon tissue of pigs in the *Lactobacillus* group tended to show increased villus length and height/crypt depth compared with that of the basal diet group (Appendix A). *Lactobacillus* did not alter muscle fibers in the longissimus dorsi muscle tissue (Appendix A). In adipose tissues, we noted a thin cell membrane covering the cytoplasmic lipid and normal nucleus, and it was pushed to one side by the lipid corner, and normal connective tissue septae were observed in fat cells. The size of the droplets was significantly reduced in the *Lactobacillus* group compared with that in the basal diet group (Appendix A). Furthermore, we did not find any unusual changes in the fat cells of the *Lactobacillus* group. Normal histology of intact hepatocytes, central vein, prominent nucleus, and sinusoidal space was observed in both groups (Appendix A).

## 4. Discussion

Probiotic Lactobacilli has gained significant interest as a potential therapy for improving human and animal health. However, the mechanistic basis underlying their beneficial properties remains unclear. In this study, we investigated the effects of dietary *Lactobacillus* on cholesterol level and liver function in Korean mini pigs, JNPs. We compared growth performance, feed efficiency, and backfat thickness in basal- and *Lactobacillus*-treated JNPs. We also evaluated immune activation responses in the liver, adipose, and colon tissues of the pigs. Our results showed no significant difference in the average daily gain, feed efficiency, and backfat thickness between the *Lactobacillus* and basal diet groups. In contrast, several probiotics, such as *Lactobacillus acidophilus*, *Lactobacillus reuteri*, *Lactobacillus gasseri*, *L. plantarum*, *L. fermentum*, *Pediococcus acidilactici*, *Lactobacillus jensenii*, and *Lactobacillus johnsonii*, have been shown to increase BW, feed conversion, feed intake, serum IgG level, and crude and digestible protein levels in pigs [18]. Recently, a comprehensive meta-analysis showed that probiotics tended to improve average daily gain and feed efficiency in swine between weaning and below 18 kg [3]. Our results in corroboration with other studies suggested that feeding probiotics have no effects on the growth performances of swine during the grower phase. 

Our findings showed that total plasma protein was increased in the *Lactobacillus* group compared with that in the basal diet group (*p* = 0.08) (Table 2). Probiotics have been reported to enhance immune activity, as indicated by significant increases in IgG levels [19]. Similarly, IgG concentration was increased in the *Lactobacillus* group compared with that in the basal diet group (Figure 1). In the present study, the average daily gain slightly decreased in the *Lactobacillus* group, which might be correlated with the reduction in the mRNA and protein expression of adipokines, including adiponectin, chemerin, and visfatin, in the adipose tissue of the *Lactobacillus* group. Moreover, the expression of PPAR-γ, the most important transcriptional modulator of adipocyte development in all types of adipose tissue, was reduced in the adipose tissue of the *Lactobacillus* group (Appendix A). Additionally, the size of fat droplets in adipose tissues was slightly reduced in the *Lactobacillus* group compared with that in the basal diet group (Appendix A).

Chemerin plays a significant role in glucose metabolism, obesity, insulin resistance, adipose tissue inflammation, and liver pathology [20]. Adiponectin is exclusively secreted from adipose tissue, and it acts as an autocrine factor in adipose tissues, promoting cell proliferation and differentiation of pre-adipocytes to mature adipocytes [21]. An important finding of this study is that dietary *Lactobacillus* supplementation significantly reduced blood cholesterol concentration as well as adiponectin and chemerin secretion from adipose tissues. In modern societies, high-fat and high-sucrose diets usually lead to hypercholesterolemia and atherosclerosis due to increased cholesterol levels [22]. Cholesterol is the principal lipid in the bile [23]. A high serum cholesterol level is a risk factor for cardiovascular diseases [24]. The *Lactobacillus* group showed lower BW gain, fat mass, and adipocyte size than the basal diet group. This efficacy of *Lactobacillus* in reducing cholesterol levels suggests that *Lactobacillus* supplementation has promising cholesterol-lowering effects. 

Blood biochemical analysis showed significantly reduced BUN (*p* = 0.08), ALP (*p* = 0.08), and GGT activity (*p* = 0.08) in the *Lactobacillus* group, which suggested that dietary supplementation with *Lactobacillus* significantly improved liver and kidney function. The maintenance of immune homeostasis by the gut microbiota in the intestine is essential [25]. The gastrointestinal tract contains many bacterial species that play a significant role in the innate immune system and trigger adaptive immune responses via various cytokines [26]. Cytokines from leukocytes or infected tissues can regulate both innate and acquired immunity [27]. TNF-α, IL-1β, IL-6, IL-12, and IFN-γ are expressed in leukocytes and all the organs or tissues involved in the immune response [28]. *Lactobacillus* spp. augment the production of immune response-related cytokines, such as TNF-α, IL-1β, IL-6, IL-12, and IFN-γ, which trigger a physiological immune response [29]. *Lactobacillus* spp. can induce the secretion of TNF-α from dendritic cells and peripheral blood mononuclear cells and increase the number of TNF-α-producing cells in the gut lamina propria [30]. For example, *L. jensenii* strongly induces the secretion of IL-10 and IFN-γ, which is associated with the beneficial effects of the immunobiotic strain [31]. In addition, Bifidobacterium longum supplementation increases the mRNA levels of IL-10 and TNF-α [32]. In the current study, the mRNA and protein expression levels of IL-1β, TNF-α, IL-6, and IFN-γ were slightly or significantly reduced in the colon tissues of the *Lactobacillus* group compared with those in the basal diet group. However, TNF-α and IFN-γ expression was increased in the liver tissue of the *Lactobacillus* group. TNF-α and IFN-γ are expressed in activated CD4+ T cells to induce an immune response [33]. In particular, IFN-γ is essential for the maturation of immune cells and regulates cellular proliferation in the intestine [34]. Many reports have claimed that *Lactobacillus* increases the number of IFN-γ-producing cells [35,36]; for example, *L. plantarum* elevates IFN-γ and IL-10 expression in the ileum mucosa of chickens [37]. Contrary to previous findings, IFN-γ levels did not change significantly in the colon tissues of the *Lactobacillus* group in this study. However, increased TNF-α and IFN-γ levels were observed in the livers of the *Lactobacillus* group compared with those in the vehicle group. Unchanged IL-1β, TNF-α, IL-6, and IFN-γ expression was observed in the adipose (Appendix A) and longissimus dorsi muscle tissues of the *Lactobacillus* group compared with that of the basal diet group. Reportedly, exposure of antigen-presenting cells to *L. jensenii* induces various cytokines, such as IL-1β, IL-6, IL-2, IL-4, IL-12, IL-10, TNFα, and IFN-γ [38]. In the present study, we noted a similar response to TNF-α and IFN-γ in the liver tissue of the *Lactobacillus* group. These results suggest that dietary supplementation with *Lactobacillus* can modulate the immune response and improve organ defense. 

TLR, which belongs to the prominent type I transmembrane receptor protein family in mammals, plays an essential role in defense mechanisms by activating innate immunity and stimulating the adaptive immune system [39]. Cells recognize the peptidoglycan of resident microorganisms mainly via NOD1 or TLR [40]. TLR4, an important member of the TLR family, is involved in various inflammatory reactions [41]. The outer membrane of bacteria is recognized by TLR4 on CD4+ T cells and activates signaling cascades involving MYD88 to induce T cell differentiation [42]. TLR4 transmits signals via MYD88-dependent or -independent pathways. As a key adaptor molecule in the TLR signaling pathway, MYD88 is widely expressed in various tissues, especially in immune and intestinal tissues, and this expression pattern is related to its role in immune functions [43]. The present study showed that the mRNA and protein levels of TLR4 and MYD88 differed in liver and colon tissues in response to *Lactobacillus*. Both TLR4 and MyD88 mRNA and proteins were significantly upregulated in the liver tissues of the *Lactobacillus* group. MYD88 expression was significantly upregulated in the colon tissue of the *Lactobacillus* group, although the expression of TLR4 was not significant. 

Previous studies showed that *L. jensenii* exposure to antigen-presenting cells does not alter the expression of TLR4 [38] and that *Lactobacillus* amylovorus blocks enterotoxigenic *Escherichia coli*-induced IL-8 and IL-1β expression by downregulating TLR4 and MYD88 signaling [44]. Interestingly, the present study showed a controversial finding: dietary supplementation with *Lactobacillus* increased TLR4 and MyD88 levels in the liver but slightly decreased TLR4 level and increased MyD88 level in colon tissues. This contradictory result may be due to the provision of only *Lactobacillus* dietary feed without stimulators, such as lipopolysaccharide or pathogenic bacteria. 

Moreover, NOD1 and NOD2 expression in the liver and colon were upregulated in the *Lactobacillus* group compared with that in the vehicle group. As previously discussed, NOD and TLR play an essential role in identifying bacteria and initiating immunological responses in the intestine. Thus, our findings suggest that NOD1 and NOD2 are involved in *Lactobacillus* recognition and immune responses, at least in colon tissues.

## 5. Conclusions

In summary, we evaluated the effects of dietary *Lactobacillus* on immune activation-related cytokines, TLR signaling, adipokines, and liver function indicators in JNPs. Our results showed that TLR4 signaling was the primary response against *Lactobacillus*, and TNF-α/IFN-γ were important immune response cytokines in the colon and liver tissues. An important finding of this study was that dietary *Lactobacillus* supplementation significantly improved liver function (as shown by reduced levels of BUN, ALP, and GGT) and reduced cholesterol levels. Taken together, our results showed that *Lactobacillus* supplementation might be beneficial against liver disorders, especially cholesterol-related disorders. Considering that previous microbiome research has been performed in rodent models, this study in monogastric mini pigs provides novel data for microbiome research. Finally, this study clearly demonstrated the potential of probiotics in positively affecting animal health. 

## Figures and Tables

**Figure 1 animals-11-02309-f001:**
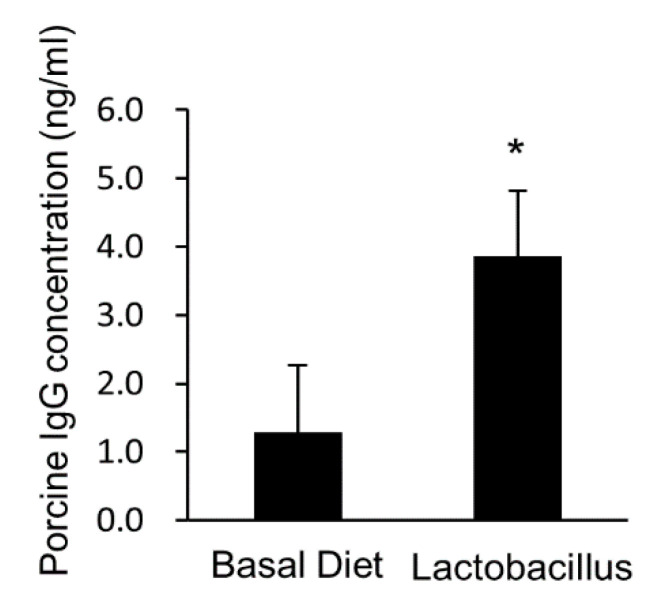
Multi-probiotic *Lactobacillus* feeding increases blood IgG level in Jeju native pigs. IgG level in the plasma of experimental pigs (*n* = 9) was quantified using an ELISA kit. The data are represented as mean ± standard error of the mean, as analyzed by two-tailed Student’s *t*-test. * *p* < 0.05, compared to the basal diet group.

**Figure 2 animals-11-02309-f002:**
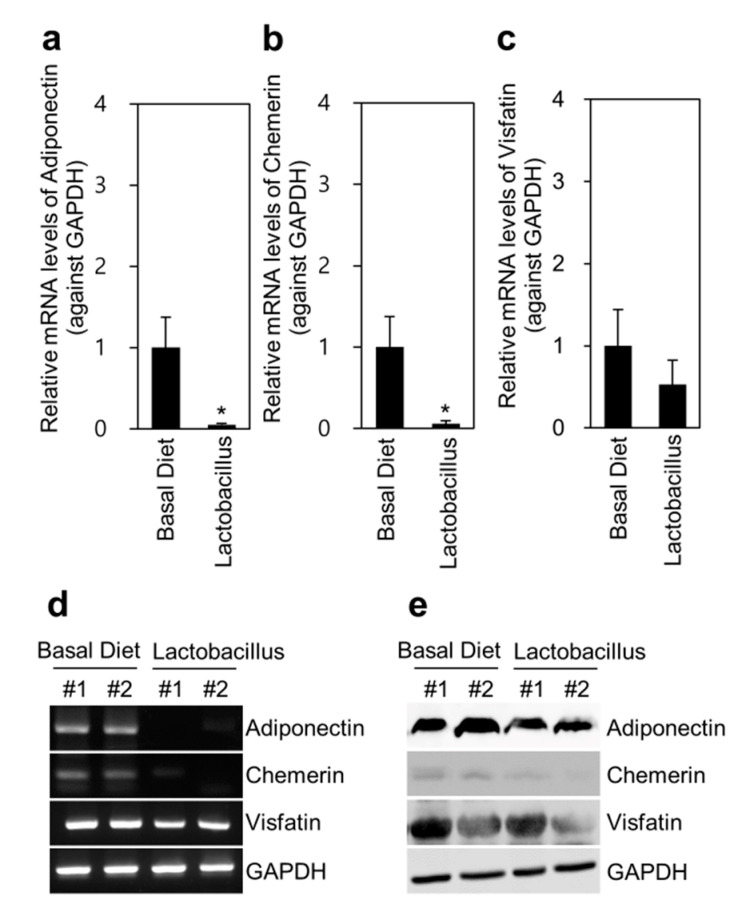
Multi-probiotic *Lactobacillus* feeding reduces adipokines in the adipose tissue of Jeju native pigs. We determined the mRNA and protein levels of adiponectin, chemerin, and visfatin in adipose tissues. The *Lactobacillus* group showed significantly reduced adiponectin and chemerin levels compared with the basal diet group. mRNA levels were determined using the SYBR green-based PCR method with targeted primers. Protein expression was detected by western blotting analysis and the chemiluminescence method. (**a**–**c**) RT-PCR quantification of adipokine levels in experimental pigs (*n* = 9). (**d**) Adipokine levels were measured by conventional PCR. (**e**) Adipokine levels were analyzed by western blotting with the respective antibodies. The data are presented as the mean ± standard error of the mean, as analyzed by two-tailed Student’s *t*-test. * *p* < 0.05, compared with the basal diet group.

**Figure 3 animals-11-02309-f003:**
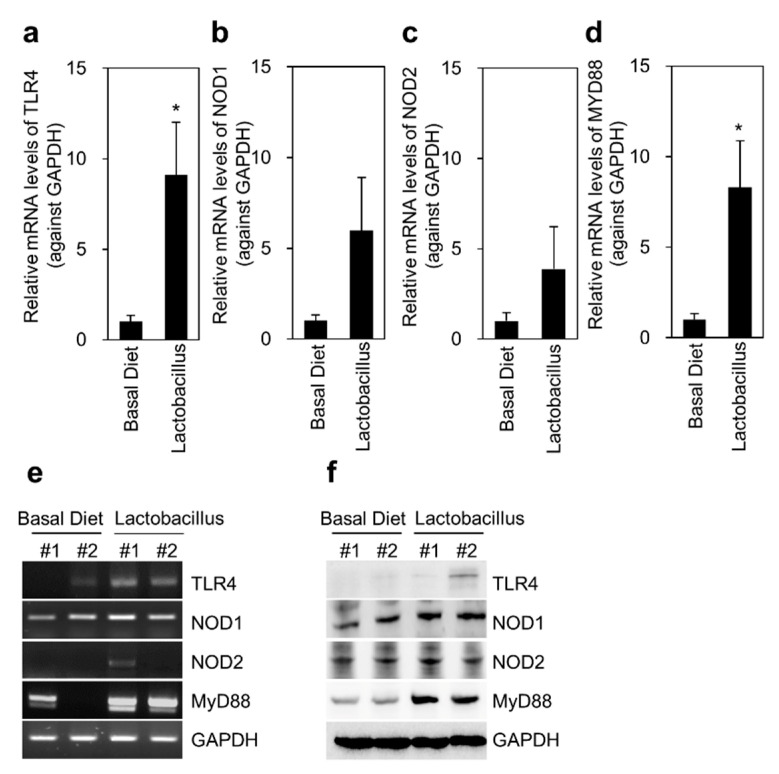
Effects of TLR4 and NOD signaling in the liver tissue of Jeju native pigs fed the multi-probiotic *Lactobacillus*. Immune activation-related cytokines were determined at both the transcriptional and translational levels using PCR and western blotting analyses, respectively. TLR4, NOD1, NOD2, and MYD88 levels were analyzed in the liver tissue of both experimental groups. (**a**–**d**) TLR4, NOD1, NOD2, and MYD88 levels were determined by qRT-PCR (*n* = 9). (**e**) TLR4, NOD1, NOD2, and MYD88 levels were determined by conventional PCR. (**f**) Immunoblot images of TLR4, NOD1, NOD2, and MYD88. The data are presented as the mean ± standard error of the mean, as analyzed by two-tailed Student’s *t*-test. * *p* < 0.05, compared with the basal diet group.

**Figure 4 animals-11-02309-f004:**
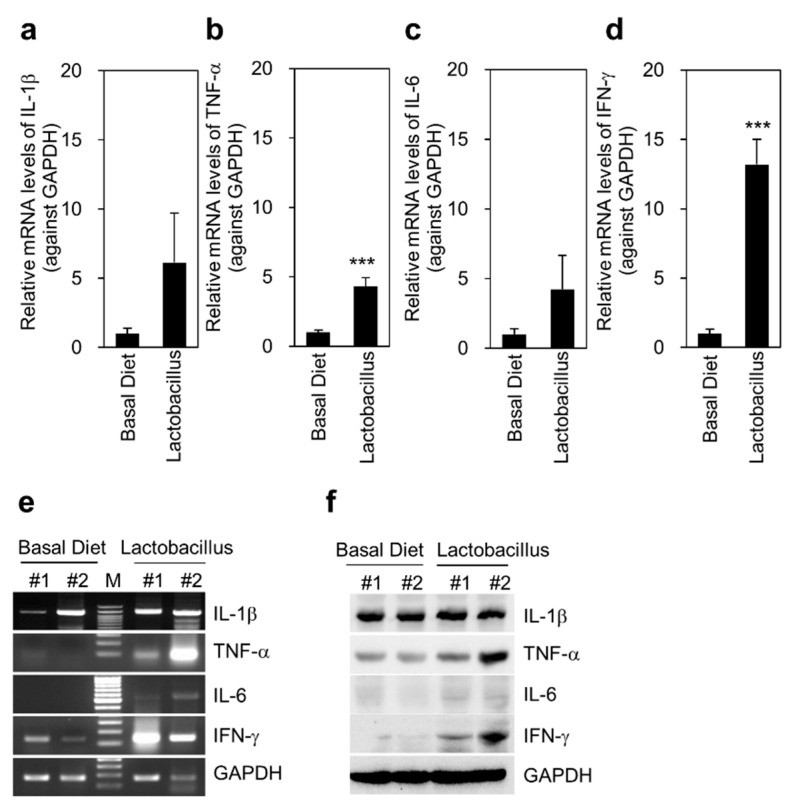
Multi-probiotic *Lactobacillus* feeding increases immune response-related cytokines in the liver tissue of Jeju native pigs. Immune activation-related cytokines were determined at both the transcriptional and translational levels using qRT-PCR and western blotting analyses, respectively. (**a**–**d**) mRNA levels of immune activation-related cytokines in liver tissues were quantified by qRT-PCR analysis (*n* = 9). (**e**) mRNA levels of immune activation-related cytokines in liver tissues were analyzed by conventional PCR. (**f**) Protein levels immune activation-related cytokines in liver tissues were determined by western blotting analysis. The data are presented as the mean ± standard error of the mean, as analyzed by two-tailed Student’s *t*-test. *** *p* < 0.001, compared with the basal diet group.

**Figure 5 animals-11-02309-f005:**
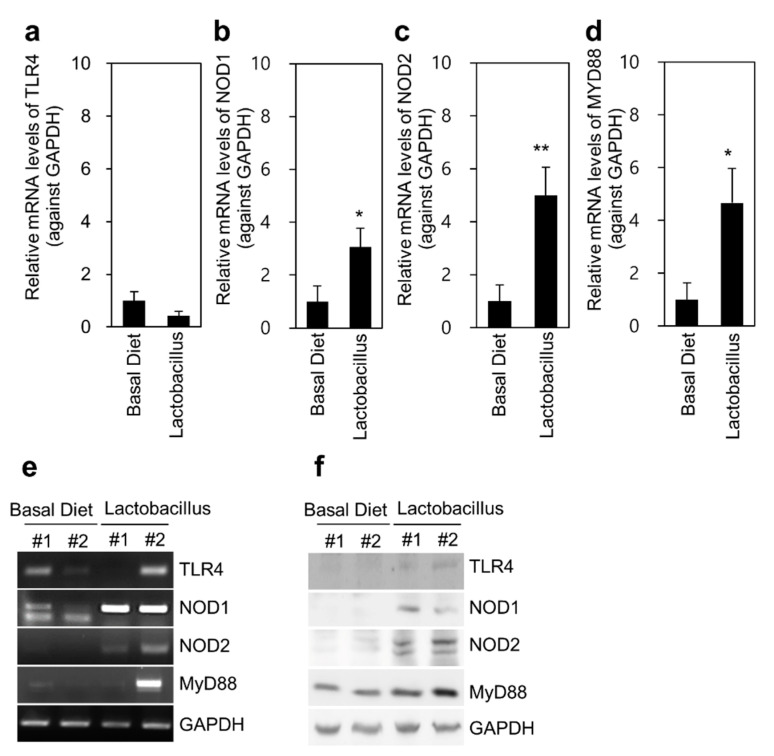
TLR4 and NOD signaling in the proximal colon tissue of Jeju native pigs fed the multi-probiotic *Lactobacillus*. TLR4, NOD1, NOD2, and MyD88 expression was determined at the mRNA and protein levels by qRT-PCR and western blotting analyses. (**a**–**d**) TLR4, NOD1, NOD2, and MYD88 expression in the proximal colon tissue was determined by qRT-PCR analysis (*n* = 9). (**e**) TLR4, NOD1, NOD2, and MYD88 expression in the colon tissue was determined by conventional PCR analysis. (**f**) TLR4, NOD1, NOD2, and MYD88 expression in the colon tissue was analyzed by western blotting. The data are presented as the mean ± standard error of the mean, as analyzed by two-tailed Student’s *t*-test. * *p* < 0.05 and ** *p* < 0.01, compared with the basal diet group.

**Table 1 animals-11-02309-t001:** Bodyweight, average daily gain, feed intake, feed efficiency, and backfat thickness in Jeju native pigs of both groups during the feeding period, 3 months.

	Basal Diet	*Lactobacillus*	SEM	*p* Value
Body weight (kg)				
Initial	32.75	32.63	2.57	0.96
Final	60.38	56.50	3.47	0.31
Average daily gain (g/d) ^†^	306.94	265.28	37.38	0.31
Feed intake (g/d)	1.60	1.60		
Feed efficiency ^§^	191.84	164.16	23.27	0.28
Backfat thickness (mm)				
Initial	7.75	8.75	0.89	0.30
Final	14.00	14.75	1.03	0.49

^†^ Animals were weighed monthly and fed the same amount of feed at 09.00 h throughout the experiment; these data are summarized as the mean value for a month of the feeding period for both diets. SEM, standard error of the mean. ^§^ Expressed as g BW gain/kg intake.

**Table 2 animals-11-02309-t002:** Blood biochemical analysis of Jeju native pigs fed the basal and multi-probiotic *Lactobacillus* diets.

	Initial				Final			
	Basal Diet	*Lactobacillus*	SEM	*p* Value	Basal Diet	*Lactobacillus*	SEM	*p* Value
Glucose (mg/dL)	112.0	116.3	3.97	0.33	102.5	96.0	11.87	0.60
Creatinine (mg/dL)	0.9	0.8	0.04	0.03 *	1.2	1.1	0.16	0.31
BUN (mg/dL) ^†^	10.3	11.8	1.06	0.21	20.0	14.5	2.60	0.08
BUN/Creatinine	11.0	14.5	1.44	0.05 *	17.0	14.3	3.47	0.46
Inorganic Phosphate (mg/dL)	9.4	9.1	0.49	0.50	8.2	7.5	0.37	0.11
Calcium (mg/dL)	11.2	10.7	0.20	0.06	10.6	10.4	0.25	0.52
Total Protein (g/dL)	7.6	8.0	0.27	0.13	8.2	9.1	0.44	0.08
Albumin (g/dL)	3.4	3.6	0.13	0.19	4.0	4.1	0.10	0.35
Globulin (g/dL)	4.2	4.5	0.16	0.14	4.1	5.0	0.45	0.11
Albumin/Globulin	0.8	0.8	0.03	0.36	1.0	0.9	0.07	0.19
ALT (U/L) ^†^	79.3	84.5	18.61	0.79	51.0	65.5	16.74	0.42
ALP (U/L) ^†^	154.5	144.5	40.20	0.81	102.0	59.8	20.50	0.08
GGT (U/L) ^†^	28.3	28.0	4.99	0.96	91.0	40.0	24.40	0.08
Total Bilirubin (mg/dL)	0.1	0.2	0.05	0.36	0.3	0.6	0.11	0.03 *
Cholesterol (mg/dL)	73.5	69.8	10.67	0.74	94.5	71.5	7.48	0.02 *
Amylase (U/L)	563.8	540.5	114.68	0.85	699.3	602.0	165.07	0.58
Lipase (U/L)	11.0	14.5	4.61	0.48	67.5	46.0	30.93	0.51

^†^ BUN, blood urea nitrogen; ALT, alanine aminotransferase; ALP, alkaline phosphatase; GGT, gamma-glutamyltransferase; SEM, standard error of the mean. Significant differences between variables in the basal diet and *Lactobacillus* groups. Comparisons were performed using independent Student’s *t*-test. * *p* < 0.05, compared with the basal diet group.

## Data Availability

Not applicable.

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
