# Peer review of "Multi-Probiotic Lactobacillus Supplementation Improves Liver Function and Reduces Cholesterol Levels in Jeju Native Pigs"

_animals, 2021, doi:10.3390/ani11082309_

Round 1

Reviewer 1 Report

The paper aims to study the role of probiotics in reducing cholesterol and liver function, which is a hot topic of research today, with real-time and hot spot, but it lacks some more favorable evidence. The relevant opinions are as follows:

L54-L61:The first paragraph is repeated with the beginning of the second paragraph

L66:The verb agrees with its subject

L172-L173:Use the correct "p" format.please use p values in a constant manner.

L329:The verb agrees with its subject.Change "have" to "has"

L402:Use the correct predicate form.Change "belong" to "belongs"

L403:Change "play"to "plays"

L425:Change "was" to "were"

L440:Scientists have been working hard to improve the shortcomings of miniature pigs. In the conclusion part, we can highlight the parts that probiotics can improve

Author Response

Reviewer #1:

The paper aims to study the role of probiotics in reducing cholesterol and liver function, which is a hot topic of research today, with real-time and hot spot, but it lacks some more favorable evidence. The relevant opinions are as follows:

Comment 1.

L54-L61:The first paragraph is repeated with the beginning of the second paragraph.

[Response]

Thank you for pointing this out. The related sentences have been revised as follows;

[Probiotics are beneficial microorganisms that compete with harmful intestinal microflora in a host [1]. They have a remarkable ability to alter the intestinal microbiota, stimulate intestinal immunity, improve resistance to diseases, reduce pathogen shedding, and improve health status [2]. Therefore, the application of probiotics on human health and farm animal production has been widely investigated [3].] (Page 2, line 53−57)

Comment 2.

L66:The verb agrees with its subject

[Response]

Thank you for your comment. The verb has been revised (page 2, line 74).

Comment 3.

L172-L173:Use the correct "p" format.please use p values in a constant manner.

[Response]

Thank you for the correction. The “p” value has been edited to the correct format (page 4, lines 171−172).

Comment 4.

L329:The verb agrees with its subject.Change "have" to "has"

[Response]

Thank you for the correction. The verb has been revised accordingly (page 10, line 334).

Comment 5.

L402:Use the correct predicate form.Change "belong" to "belongs"

[Response]

Thank you for the correction. The word has been revised accordingly (page 11, line 411).

Comment 6.

L403:Change "play"to "plays".

[Response]

Thank you for the correction. The word has been revised accordingly (page 11, line 412).

Comment 7.

L425:Change "was" to "were"

[Response]

Thank you for the correction. The word has been revised accordingly (page 12, line 434).

Comment 8.

L440:Scientists have been working hard to improve the shortcomings of miniature pigs. In the conclusion part, we can highlight the parts that probiotics can improve

[Response]

Thank you for your pertinent suggestion. Accordingly, the beneficial effects of probiotics has been highlighted in the conclusion section of the revised manuscript (page 12, lines 449−451).

Reviewer 2 Report

This is an interesting trial that investigated the effects of supplementation with Lactobacillus in Jeju Native piglets’ diet on the liver function and cholesterol. The findings indicated dietary Lactobacillus supplementation improved liver function and reduced cholesterol levels. Lactobacillus supplementation may be applied to treat metabolic disorders of the liver, especially cholesterol-related disorders. However, in trial, the no significant effects on performance of piglets due to the supplementation of Lactobacillus. Some detailed questions need a precise expression in the manuscript.

  1. in the 2.1 experimental procedure, the authors need to give a detailed expression to the source of multi-Lactobacillus, and also the author mentioned that they took oligosaccharide as a substrate in the multi-bacteria preparation. The bacteria preparation products used in the experiment were purchased from some companies or cultured in the laboratory? What’s the type of the oligosaccharide? Yeast cell wall or other sources oligosaccharide? Because the oligosaccharides also are important prebiotic and exert multi-bioactivities to host. The authors must give a detailed description to the experimental materials.

  1. in this trial, the dietary Lactobacillus supplementation improved liver function and reduced cholesterol levels, while the performances (Average daily gain, feed intake, feed efficiency, backfat thickness, and body weight) didn’t change. The authors should give a expression in discussion that whether it is recommendable to supplement the Lactobacillus preparation in piglet’s diet for the aim of improve the production level in JNPs industry?

Author Response

Reviewer #2:

This is an interesting trial that investigated the effects of supplementation with Lactobacillus in Jeju Native piglets’ diet on the liver function and cholesterol. The findings indicated dietary Lactobacillus supplementation improved liver function and reduced cholesterol levels. Lactobacillus supplementation may be applied to treat metabolic disorders of the liver, especially cholesterol-related disorders. However, in trial, the no significant effects on performance of piglets due to the supplementation of Lactobacillus. Some detailed questions need a precise expression in the manuscript.

Comment 1.

  1. in the 2.1 experimental procedure, the authors need to give a detailed expression to the source of multi-Lactobacillus, and also the author mentioned that they took oligosaccharide as a substrate in the multi-bacteria preparation. The bacteria preparation products used in the experiment were purchased from some companies or cultured in the laboratory? What’s the type of the oligosaccharide? Yeast cell wall or other sources oligosaccharide? Because the oligosaccharides also are important prebiotic and exert multi-bioactivities to host. The authors must give a detailed description to the experimental materials.

[Response]

Thank you for your important comment. First, the multi-probiotic Lactobacillus used in this study was purchased from a company. The strain name and company information has been provided in the Material and Methods section of the revised manuscript (page 3, lines 103−107).

Second, the fructo-oligosaccharide used in the formulation of probiotics was gotten from Saccharum barberi. The origin of the fructo-oligosaccharide used in this study has been added to the Material and Methods section of the revised manuscript (page 3, lines 107~108).

Comment 2.

  1. in this trial, the dietary Lactobacillus supplementation improved liver function and reduced cholesterol levels, while the performances (Average daily gain, feed intake, feed efficiency, backfat thickness, and body weight) didn’t change. The authors should give a expression in discussion that whether it is recommendable to supplement the Lactobacillus preparation in piglet’s diet for the aim of improve the production level in JNPs industry?

[Response]

Following the reviewer’s comment, the following revision has been made to the revised manuscript to address the concerns raised (page 10, lines 345~349);

[Recently, a comprehensive meta-analysis showed that probiotics tended to improve average daily gain and feed efficiency in swine between weaning and below 18 kg [3]. Our results in corroboration with other studies suggested that feeding probiotics have no effects on the growth performances of swine during the grower phase.]

Reviewer 3 Report

Dear authors,

Thank you for your interesting work. I have only a few comments to suggest:

Line 61-61 : This is not a mechanism

Line 84-85: maybe you could add more information about the liver as a metabolic and excretory organ

Line 72: via "the release"

line 89: Please revise the phrase "develop JNPs industry" 

The introduction section could be improved to provide more clear and concise information, with natural flow and connectivity.

Author Response

Reviewer #3:

Dear authors,

Thank you for your interesting work. I have only a few comments to suggest:

.

Comment 1.

Line 61-61 : This is not a mechanism

 [Response]

Thank you for your comment. The related portion has been revised (page 2, line 70).

Comment 2.

Line 84-85: maybe you could add more information about the liver as a metabolic and excretory organ

[Response]

Thank you for your comment. More information about the liver function has been provided accordingly (page 2, line 66−69).

Comment 3.

Line 72: via "the release"

[Response]

Thank you for your comment. The term “the release” has been added to the revised manuscript (page 2, lines 80~81).

Comment 4.

line 89: Please revise the phrase "develop JNPs industry" 

[Response]

Thank you for your comment. The phrase has been revised in the updated manuscript (page 2, line 86).

Comment 5.

The introduction section could be improved to provide more clear and concise information, with natural flow and connectivity. 

[Response]

Thank you for your important suggestion. The introduction section has been revised accordingly (page 2, line 53−94).